# Non-1st seizure was less severe than 1st seizure with non-urgent level among suspected seizures transferred by ambulance

**Yotaro Asano[1], Ayataka Fujimoto[2]\*, Keisuke Hatano[2], Keishiro Sato[2], Takahiro Atsumi[3], Hideo Enoki[2], Tohru Okanishi[4]**

1 Hamamatsu University School of Medicine, Shizuoka, Japan, 2 Comprehensive Epilepsy Center, Seirei Hamamatsu General Hospital, Shizuoka, Japan, 3 Department of Emergency Medicine, Seirei Hamamatsu General Hospital, Shizuoka, Japan, 4 Division of Child Neurology, Department of Brain and Neurosciences, Faculty of Medicine, Tottori University, Yonago, Japan

\* afujimotoscienceacademy@gmail.com

## Abstract

### Background

To prioritize emergency medical calls for ambulance transport for patients with suspected seizures, information about whether the event is their 1st or non-1st seizure is important. However, little is known about the difference between 1st and non-1st seizures in terms of severity. We hypothesized that patients transferred multiple times ($\geq$2 times) would represent a milder scenario than patients on their first transfer. The purpose of this study was to compare patients with suspected seizures on 1st transfer by ambulance and patients who had been transferred $\geq$2 times.

### Methods

We statistically compared severity of suspected seizures between two groups of patients with suspected seizures transferred between December 2014 and November 2019 (before the coronavirus disease 2019 pandemic) to our facility by ambulance for either the first time (1st Group) or at least the second time (Non-1st Group). Severity categories were defined as: Level 1 = life-threatening; Level 2 = emergent, needing admission to the intensive care unit; Level 3 = urgent, needing admission to a hospital general ward; Level 4 = less urgent, needing intervention but not hospitalization; and Level 5 = non-urgent, not needing intervention.

### Results

Among 5996 patients with suspected seizures conveyed to the emergency department by ambulance a total of 14,263 times during the study period, 1222 times (8.6%) and 636 patients (11%) met the criteria. Severity grade of suspected seizures ranged from 1 to 5 (median, 4; interquartile range, 3–4) for the 1st Group and from 1 to 5 (median, 5; interquartile range, 4–5) for the Non-1st Group. Most severe grade ranged from 1 to 5 (median, 4; interquartile range, 4–5) for the Non-1st Group. Severity grade differed significantly between

**Data Availability Statement:** All relevant data are within the manuscript.

**Funding:** The authors received no specific funding for this work.

**Competing interests:** The authors have declared that no competing interests exist.

groups (p < 0.001, Mann–Whitney U-test). Uni- and multivariate logistic regression tests also suggested a significant difference (p < 0.001) in severity grades.

## Conclusion

In direct comparisons, grade of suspected seizure severity was lower in the Non-1st Group than in the 1st Group.

## 1. Introduction

Many guidelines and instructive movies do not include an emergency medical call as part of basic first aid in cases of seizure. For example, the websites of the Centers for Disease Control and Prevention (CDC) (https://www.cdc.gov/epilepsy/about/first-aid.htm), International League Against Epilepsy ILAE) (https://www.ilae.org/patient-care/for-persons-with-epilepsy-and-caregivers/first-aid-during-a-seizure), Epilepsy Action Australia (https://www.epilepsy.org.au/about-epilepsy/first-aid/), Japan Division, and International Bureau for Epilepsy (https://www.jea-net.jp/epilepsy/spasm) do not recommend an emergency medical call, but basic first aid is the first priority. The Swiss League Against Epilepsy (https://www.youtube.com/watch?v=-XcWmksrEBk), the seizure first aid poster of the Epilepsy Foundation (https://www.epilepsy.com/recognition/first-aid-resources) and Epilepsy Action Australia all provide educational instructions about first aid for epileptic seizures. According to those posters and movies, basic first aid should be provided as follows: 1) stay with the person until they are awake and alert after the seizure [time the seizure, remain calm, and check for medical identification (ID)]; 2) keep the person safe (move or guide them away from harm); 3) turn the person onto their side and then, finally, consider calling an ambulance. According to these guidelines, a patient showing: 1) seizure lasting <5 min; 2) self-limited seizures; and 3) habitual stereotypical seizures, not the first seizure, in most cases only needs basic first aid, not ambulance transfer or a visit to the emergency department (ED) [1–3]. However, inappropriate or unnecessary ambulance calls are a problem in the real world [4, 5] and have long been discussed [1, 6]. This issue remains topical [7, 8].

The reason for timing seizures is that convulsive seizures lasting >5 min can be considered status epilepticus (SE) [9], in which the risk of neuronal injury or neuronal death is greatly increased [9]. The reason for noting whether a seizure event represents the first seizure is that an acute symptomatic seizure (ASS) might be caused by an irreversible condition, such as stroke, head trauma, or infection [10–12]. Medical IDs such as a bracelet or tag might therefore help decide whether an event is the first seizure or a habitual stereotypical epileptic seizure in a patient with established epilepsy.

When considering an ambulance call, the guideline can be simplified into: 1) does the seizure event represent SE or not; 2) does the seizure event represent a first or non-first seizure; and 3) does the individual have concomitant symptoms such as physical injuries, fever, or non-stereotypical seizures. The presence of SE and/or concomitant symptoms can understandably be considered to warrant a call to emergency services. However, whether the first seizure might represent a more severe condition than non-first seizure is unclear because direct comparisons of severity between first and non-first seizures transferred by ambulance have not been conducted.

It is important to note that seizure severity may not be the main factor that should drive the decision to call an ambulance. Even if a first seizure was not severe, the fact that there may be

an acute cause for the first event requiring urgent treatment suggests that first seizures should be worked up urgently [13].

This does not apply for the second seizure and the authors are, therefore, looking at the severity of the actual event.

Little is known about actual differences in severity between first and non-first seizures to decide on ambulance transfers. Therefore, among patients transferred by ambulance with possible central nervous system (CNS) disorders that require differentiation from epilepsy (suspected seizure), whether non-first seizures are really milder than first seizures, how severe first and non-first seizures are, and the extent to which we should be aware of differences in severity risk between first and non-first seizure are issues that remain unclear.

In this study, we hypothesized that patients who were not on their first transfer (≥2 trips) would have milder settings than patients on their first transfer. Previous studies have examined the relationship between ambulance transfer and medical costs [8, 14] and seizure characteristics in the ED [15, 16]. However, only one case-control study about predictors of ED visits [17] has been performed in terms of differences in severity between first and non-first ambulance transfers.

The purpose of this study was to compare severity grades of patients with suspected seizures between those on their first transfer by ambulance and those transferred ≥2 times.

## 2. Methods

### 2.1. Study design and ethics approval

This retrospective observational study compared two groups from a single tertiary center. The design and analysis of this study were conducted in accordance with the Strengthening the Reporting of Observational Studies in Epidemiology (STROBE) statement checklist [18].

All participants provided written or verbal informed consent prior to inclusion in the study. The verbal consent was chart-recorded in the presence of an independent witness. Written informed consent for publication of data pertaining to participants under the age of 18 years was obtained from the patients' guardians. In addition, in cases of severe patients or those who have unfortunately passed away, we have obtained consent from their caregivers. All data were securely and confidentially stored in line with ethical requirements, and data presented below have been anonymized to protect participants' identities. The ethics committee at Seirei Hamamatsu General Hospital approved the protocol for this study (approval no. 4079), which was performed in accordance with the principles of the Declaration of Helsinki.

### 2.2. Settings

We retrospectively reviewed our electronic medical records and identified patients transferred to the ED at Seirei Hamamatsu General Hospital, Shizuoka, Japan for issues considered related to CNS disorder between December 2014 and November 2019 (before the coronavirus disease 2019 pandemic).

### 2.3. Participants

Among those transferred due to possible CNS disorders, we included patients: 1) who were considered in need of attention from an epilepsy specialist among ED physicians and other general neurological physicians; 2) with epilepsy who had previously seen an epilepsy specialist in our epilepsy center; or 3) with psychiatric symptoms including psychogenic non-epileptic seizure (PNES), because these disorders need to be differentiated from true epileptic seizures or other CNS disorders. In our institution, patients transferred to the ED due to suspected

epileptic disorders are sent by ED physicians to general neurological physicians. Even if the patient is first examined by ED physicians, and/or general neurological physicians and are suspected to have an epilepsy disorder, patients are referred to epilepsy specialists through the neurological physicians.

We included not only patients with tonic-clonic seizures, but also patients with other seizure types or mimics (PNES and syncope) disorders. Exclusion criteria were: 1) patients with epilepsy who were being transferred because of non-CNS disorders such as gastrointestinal symptoms, high-energy trauma, etc.; 2) occasions that were unrelated to CNS disorder, such as gastrointestinal symptoms, high-energy trauma, etc.; 3) patients who had apparent non-epileptic CNS disorders such as subarachnoid or intracranial hemorrhage, chronic subdural hemorrhage, CNS infection, etc.; or 4) patients who had already been diagnosed with established epilepsy at other facilities and first transfer to our facility. Seizures that led to systemic critical conditions or psychiatric symptoms, including suicide ideation, were not excluded because these symptoms require differentiation from epileptic seizures.

We statistically compared two groups of participants: those on their first transfer to our facility by ambulance (1st Group); and those who had been transferred by ambulance two or more (Non-1st Group).

## 2.4. Data sources

We reviewed: 1) number of transportations; 2) grade of severity at the time of transportation (see the explanation of severity grading in Section 2.5.1); 3) age at time of transportation; and 4) sex.

**2.4.1. Primary outcome measurement.** We statistically compared severity grade between the two groups (1st Group vs. Non-1st Group) using the Mann–Whitney test, uni- and multivariate logistic regression analysis, and the chi-squared test. If Non-1st Group patients were transferred two or more times, we conducted comparisons using the most severe grade from among those transfers, rather than mean or median, in uni- and multivariate logistic regression analyses. The reason for taking the most severe grade was that this study was comparing severity, so avoiding the influence of the milder grade that would have resulted from averaging or use of the median was considered ideal.

**2.4.2. Secondary outcome measurements.** We reviewed etiology as determined after ambulance transportation (such as epilepsy or PNES) between groups.

## 2.5. Variables

**2.5.1. Definition of severity grade.** Based on worldwide ambulance response categories [19, 20], we categorized the severity of suspected seizures from life-threatening to non-urgent. Categories were: Level 1 = life threatening, with need for resuscitation; Level 2 = emergent, with need for admission to the intensive care unit; Level 3 = urgent, with need for admission to a hospital general ward; Level 4 = less urgent, with need for intervention but not hospitalization; and Level 5 = non-urgent, with no need for intervention.

**2.5.2. Etiology as determined after ambulance transportation.** We categorized etiologies as determined after ambulance transportation as: SE; possible systemic disorder such as febrile seizure (FS); definitive epileptic seizure in which a patient with an established diagnosis of epilepsy showed habitual stereotypical seizures; and PNES as defined at the ED based on laboratory test results such as no evidence of acidosis, normal creatinine kinase, prolactin and lactate levels, and atypical symptoms or later diagnosis by epilepsy specialists. Patients with PNES were finally diagnosed with long-term video-electroencephalography if needed [21].For

the Non-1st Group, patients with epilepsy who were sometimes conveyed to the ED for epileptic seizures or PNES were included as PNES for this analysis.

### 2.6. Bias

When enrolling participants, we used our electronic database. For data collection, hospital staff not involved in this study collected data from the participants. The search criteria "patients who were transported to our ED between December 2014 and November 2019" and "patients who were seen by epilepsy specialists between December 2014 and November 2019" were used to select participants, age at the time of transportation, sex, hospital ID number, and date of birth were then collected. For the diagnosis of epilepsy, physicians who were both certified as epileptologists by the Japan Epilepsy Society and certified in electroencephalography by the Japanese Society of Clinical Neurophysiology (A.F., K.S., and H.E.) diagnosed the epilepsy in each patient.

Since Japan has a universal health insurance system with free access and low cost [22], people in Japan can use an ambulance without extra fees, representing a situation that differs from some other countries.

### 2.7. Statistical analysis

Values of p < 0.05 were considered indicative of significant differences in all analyses. The Mann–Whitney U-test was used to compare each group of nonparametric pairs, but not the paired t test, since testing showed non-normal distributions. Comparisons between the two groups were also made using uni- and multivariate logistic regression tests. We also used the chi-squared test to compare severity grades between the two groups. All statistical analyses were performed using Sigma Plot version 14.5 software (Systat Software, San Jose, CA, USA).

## 3. Results

### 3.1. Main findings

Among the 5996 patients with suspected seizures conveyed to the ED by ambulance in a total of 14,263 trips during the study period, 1222 trips (8.6%) and 636 patients (11%) met the criteria. Severity of suspected seizures ranged from grade 1 to 5 (median, 4; interquartile range, 3–4) for the 1st Group and from 1 to 5 (median, 5; interquartile range, 4–5) for the Non-1st Group. Severity grade differed significantly between groups.

### 3.2. Participants

Of the 1222 trips, 636 were analyzed in this study, because we included only the most severe of multiple transports for the same patient. Age ranged from 0 to 91 years (mean, 22.3 years; median, 16 years; standard deviation, 19.7; interquartile range, 8–32 years).

### 3.3. Clinical information and descriptive data

Among the 636 patients, the 1st Group included 390 patients (210 males, 180 females; 390 trips) and the Non-1st Group included 246 patients (133 males, 113 females; 832 trips). The other 5360 patients and 13,041 trips were excluded due to non-epileptic settings, such as patients with epilepsy who had gastrointestinal disease, etc. (Fig 1).

Age in the 1st Group ranged from 0 to 87 years (mean, 21.8 years; median, 16 years; standard deviation, 19.5; interquartile range, 8–31 years). Age in the Non-1st Group ranged from 0 to 91 years (mean, 23.1; median, 17 years; standard deviation, 20.3 years; interquartile range, 8–36 years). No significant difference in age was seen between the 1st and Non-1st Groups

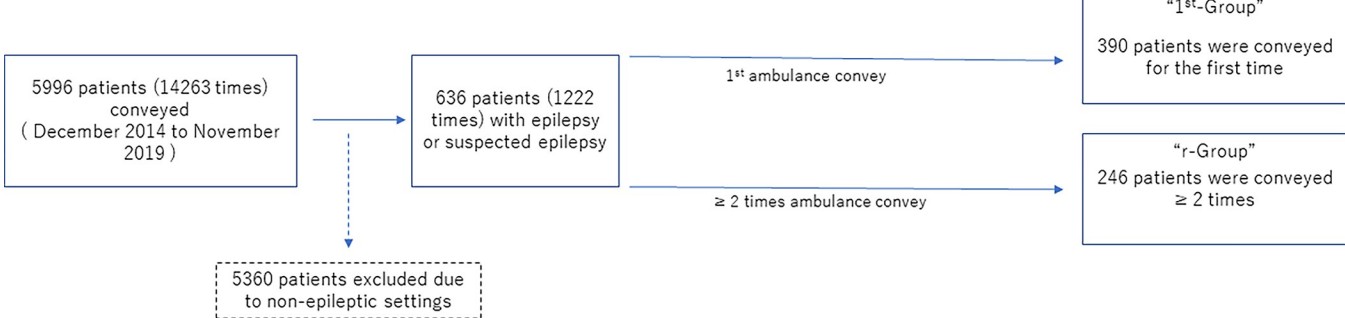

**Fig 1. Inclusion and exclusion of patients and ambulance transfers.** Among a total 14,263 trips and 5996 patients conveyed to the emergency department with symptoms related to epilepsy or suspected seizures, 1222 trips (8.6%) and 636 patients (11%) met the study criteria.

(p = 0.37). Likewise, no significant difference in sex was seen between the 1ˢᵗ Group and the Non-1ˢᵗ Group (p = 0.39). The number of ambulance trips in the Non-1ˢᵗ Group ranged from 2 to 77 (median, 3; interquartile range, 2–4) (Table 1).

### 3.4. Outcome data

**3.4.1. Primary outcome data.** Severity grade for the 1ˢᵗ Group ranged from 1 to 5 (median, 4; interquartile range, 3–4). Severity grade for the Non-1ˢᵗ Group in total also ranged from 2 to 5 (median, 5; interquartile range, 4–5). Most severe grade for the Non-1ˢᵗ Group ranged from 1 to 5 (median, 4; interquartile range, 4–5). Severity grade differed significantly between the two groups (p < 0.001) (Tables 2 and 3). In terms of severity grade distributions in the 1ˢᵗ Group and Non-1ˢᵗ Group, each grade of severity showed significant differences in grades 3, 4 and 5 ($\chi^2$ = 92.7, p < 0.01). Grade 3 was significantly more frequent in the 1ˢᵗ Group than in the Non-1ˢᵗ Group, and grades 4 and 5 were significantly more frequent in the Non-1ˢᵗ Group than in the 1ˢᵗ Group. However, no significant differences in grades 1 or 2 were seen between groups (Table 4).

**3.4.2. Secondary outcome data.** In the 1ˢᵗ Group, the etiology as determined after transport was SE in 8 patients (2%), possible systemic disorders such as ASS in 50 patients (13%), FS in 33 patients (8.5%), definitive epileptic seizure in which patients with an established diagnosis of epilepsy showed habitual stereotypical seizures in 263 patients (67%), and PNES diagnosed at ED based on laboratory tests such as no evidence of acidosis, normal creatinine kinase, prolactin and lactate levels, and atypical symptoms or later diagnosis by epilepsy specialists in 36 patients (9%).

In the Non-1ˢᵗ Group, the reason for transport was SE in 8 patients (3%), possible systemic disorders such as ASS in 23 patients (9%), FS in 20 patients (8%), definitive epileptic seizure in which patients with an established diagnosis of epilepsy showed their habitual stereotypical

**Table 1. Clinical information.**

|  | 1st Group | Non-1st Group | *P*-value |
|---|---|---|---|
| Number of patients | 390 (61.3%) | 246 (38.7%) | n/a |
| Sex (female:male) | 180:210 | 113:133 | 0.39 |
| Age (median, range, inter-quartile range) | 16, 0–87, 8–31 | 17, 0–91, 8–36 | 0.37 |
| Number of trips (median, range, inter-quartile range) | 390 (n/a) | 832 (3, 2–77, 2–4) | n/a |

Abbreviations: n/a, not available.

**Table 2. Severity grade in 1st group and Non-1st group.**

| | Most severe grade among each trip | Every trip (median, range, inter-quartile range) |
|---|---|---|
| 1st Group (median, range, inter-quartile range) | 390 times (4.0, 1–5, 3–4) | |
| Non-1st Group (median, range, inter-quartile range) | 246 times (4.0, 1–5, 4–5) | 832 times (5.0, 2–5, 4–5) |
| *P*-value | < 0.001* | < 0.001* |

\* Statistically signicicant, Mann-Whitney rank-sum test.

Severity grade: Level 1 = life threatening with need for resuscitation; Level 2 = emergent with need for admission to intensive care unit; Level 3 = urgent with need for admission to hospital general ward; Level 4 = less urgent with need for intervention but not hospitalization; Level 5 = not urgent with no need for intervention.

seizures in 161 patients (65%), and PNES in 34 patients (14%). No significant difference between the 1st and Non-1st Groups was seen in terms of etiology as determined after ambulance transport (p = 0.173) (Table 5).

## 4. Discussion

### 4.1. Key results

In the direct comparison, the Non-1st Group showed less severe grade than the 1st Group. Around 95% of the Non-1st Group was classified as grade 4 or 5, with about 5% potentially being severe grade. After excluding SE, all remaining reasons for ambulance transfer showed a mean severity grade of 4.2–4.7 in the Non-1st Group.

**4.1.1. Primary outcome.** In direct comparisons, the Non-1st Group showed less severe grade than the 1st Group. This finding may support the guidelines, posters and movies that say there is no need for an emergency call, but basic first aid should be given priority, if the individual is not experiencing their first seizure. Considering the distribution of severity grades, the more severe grades 1 and 2 made up a small proportion of cases (about 5%), but grades 3–5 included a larger number (about 95%) in both groups. Therefore, in cases of suspected seizure, we can expect that: 1) patients with a non-1st seizure will likely be less severe than those with a 1st seizure; 2) about 95% of the 1st Group will be more than grade 3; 3) about 95% of the Non-1st-Group will be more than grade 4 level; and 4) about 5% of cases might be severe grade —which is worth knowing—and ambulance transfer could be justified. In this study, most trips by the Non-1st Group were for grades 4 and 5, and thus did not require admission to hospital with only 0.2% of trips requiring resuscitation and 3% requiring admission to the intensive care unit.

**Table 3. Statistical analyses of predictors for 1st group vs. Non-1st group.**

| | Multivariate | | | Univariate | | |
|---|---|---|---|---|---|---|
| | Coefficient | Standard error | *P*-value | Coefficient | Standard error | *P*-value |
| Age | 0.009 | 0.005 | 0.055 | 0.0001 | 0.03 | 0.41 |
| Sex | -0.11 | 0.173 | 0.51 | -0.03 | 0.04 | 0.39 |
| Reason for trip | -0.08 | 0.11 | 0.48 | 0.02 | 0.02 | 0.30 |
| Severity grade | 0.82 | 0.11 | <0.001* | 0.16 | 0.02 | <0.001* |

\* Statistically significant; uni- and multivariate logistic regression analysis.

**Table 4. Severity grade distributions in 1st and Not-1st Groups.**

| Severity grade | 1 | 2 | 3 | 4 | 5 |
|---|---|---|---|---|---|
| 1st Group (390 patients) | 3 (1%) | 18 (4%) | 167 (43%)*▲ | 116 (30%)*▽ | 86 (22%)*▽ |
| Non-1st Group (246 patients) | 1 (0.8%) | 11 (4%) | 20 (8%)*▽ | 106 (43%)*▲ | 108 (43%)*▲ |
| Non-1st Group (all 832 trips) | 2 (0.2%) | 27 (3%) | 45 (5%) | 241 (29%) | 517 (62%) |

* Statistically significant: ▲, significantly larger; ▽, significantly smaller.

Since ER physicians do not perform the general workup for epilepsy in our facility, the 1st Group was referred to an epilepsy specialist when a general workup for epilepsy was needed.

Since we did not study whether specific trips were unnecessary in this study, we cannot conclude that most trips for the Non-1st Group were unnecessary. However, the collected data revealed that most trips for the Non-1st Group did not require admission to hospital. Therefore, patients with epilepsy or suspected seizure can be discouraged from visiting the ED unless complications have been recognized [15, 23, 24]. However, those epidemiologic studies were mostly demographically reported, but did not include statistical analyses [5, 25, 26]. From this perspective, the present study provides some statistical support for the earlier demographic reports.

**4.1.2. Secondary outcomes.** In terms of symptoms, patients with etiology of SE, possible systemic disorder such as ASS, or FS would naturally require ambulance transfer because of the need for medical treatment. In addition, since 9% of the Non-1st Group also showed ASS, caution is warranted regarding the fact that even patients with established epilepsy who have been repeatedly transferred by ambulance might have ASS. However, all reasons except for SE showed a mean grade of 4.2–4.7, representing a less-urgent to non-urgent level, and about 65% patients with epilepsy, which was slightly larger than a previous study with 60% [27] and 14% of PNES, larger than a previous study with 5% [2, 28] in the Non-1st Group (Table 5). This means that patients with epilepsy or PNES, representing about 80%, achieved no benefit from visiting the ED by ambulance. This result is supported by a study from the National Audit of Seizure Management in Hospitals [29], which found that ED visits for patients with epilepsy achieved few improvements in epilepsy management [30]. On the one hand, some reports have found almost no benefit from visiting the ED by ambulance for patients with epilepsy, due to a lack of medical need and the patients with epilepsy themselves knowing the disadvantages of visiting the ED [30, 31]. On the other hand, only 17% of patients with epilepsy reportedly know of the lack of need for ambulance transfer for habitual stereotypical seizures [32].

**Table 5. Etiology as defined after ambulance trip in 1st and non-1st groups.**

| Etiology | 1st Group (390 patients) Severity grade (median, range, inter-quartile range) | Non-1st Group (at time of most severe grade) Severity grade (median, range, inter-quartile range) |
|---|---|---|
| Status epilepticus | 8 (2%) (2.0, 2–5, 2–3) | 8 (3%) (3.0, 2–4, 2–3.75) |
| Acute symptomatic seizures | 49 (13%) (3.0, 1–5, 3–5) | 23 (9%) (5.0, 2–5, 3–5) |
| Febrile seizures | 33 (8.5%) (4.0, 3–5, 3–5) | 20 (8%) (5.0, 3–5, 4.25–5) |
| Definitive epileptic seizures | 263 (67%) (4.0, 1–5, 3–4) | 161 (65%) (4.0, 2–5, 4–5) |
| Psychogenic non-epileptic seizures | 37 (9%) (4.0, 2–5, 3–4.5) | 34 (14%) (5.0, 1–5, 4–5) |

**Table 6. Algorithm for seizure 1st aid.**

| Basic 1st aid | Point 1 | Stay with individual until they are awake and alert after the seizure | Ascertain SE/1st-seizure by: A) timing the seizure ($\geq$5 min); B) checking for medical ID (1st seizure or non-1st seizure by established epilepsy) |
|---|---|---|---|
| | Point 2 | Keep individual safe | |
| | Point 3 | Turn individual onto their side if they are not awake and aware. | |
| Consider ambulance call | Severity grade (median, range, inter-quartile range) | 1st seizure (4.0, 1–5, 3–4) | Non-1st seizure (5.0, 1–5, 4–5) |
| | SE | 2% | 3% |
| | Other concomitant symptoms, including ASS, FS | 21.50% | 17% |
| | Epileptic seizure or PNES | 76% | 79% |

Abbreviations: ASS, acute symptomatic seizures; FS, febrile seizures; ID, identification; PNES, psychogenic non-epileptic seizures; SD, standard deviation; SE, status epilepticus.

Severity grade: Level 1 = life threatening with need for resuscitation; Level 2 = emergent with need for admission to intensive care unit; Level 3 = urgent with need for admission to hospital general ward; Level 4 = less urgent with need for intervention but not hospitalization; Level 5 = non-urgent with no need for intervention.

## 4.2. Interpretation

From this study, the severity on arrival by ambulance for patients with suspected seizures and $\geq$ 2 transfers is 95% likely to be less-urgent or non-urgent. Among these, as 65% of cases involved established epilepsy and 14% involved PNES, knowing the medical history in terms of the number of previous ambulance transfers could be important.

The instructional poster from the Epilepsy Foundation (https://www.epilepsy.com/recognition/first-aid-resources) could be modified to consider an algorithm for first aid for suspected seizures (Table 6). If we as epileptologists and ED physicians can use this algorithm to educate patients with epilepsy or PNES, unnecessary ambulance use might be able to be reduced. We intend to study this hypothesis prospectively in future work.

## 4.3. Limitations

This study did not investigate the effect on seizure occurrence of treatment changes or provocative factors such as stress, sleep deprivation, or alcohol, although these factors have been reported to aggravate seizures [33]. Studies that include these factors may be useful.

This was a single-center study of a tertiary hospital in a suburban area of Japan. In addition to this regional difference, the Japanese health insurance system also has free access and low medical costs, so not all findings might be applicable to other regions. Multi-center involvement and international studies are therefore warranted.

## 4.4. Generalizability

Even with the above-mentioned limitations of this study, many ambulance transfers for individuals with suspected seizures and a history of previous transfers are likely to involve non-severe conditions. The present findings support previous reports that ambulance transport is often unnecessary [2, 5, 8, 26, 28, 34, 35] and add weight to consideration of other pathways for patients with suspected seizures [4, 29]. To reduce unnecessary use of ambulances, in addition to education for patients, caregivers, school and work environments [17, 36, 37], or remote systems using telemetry [38, 39], providing a simplified version of Table 6 and graphic instructions as a card that can be carried [40] might be useful for patients with epileptic seizures.

## 5. Conclusion

In the direct comparison, the Non-1st Group showed less severe grade than the 1st Group. Around 95% of the Non-1st Group was more than grade 4, with about 5% potentially being severe grade. After excluding SE, all the remaining reasons for ambulance transfer showed a mean severity grade of 4.2–4.7 in the Non-1st Group.

## Author Contributions

**Data curation:** Yotaro Asano, Ayataka Fujimoto, Keishiro Sato, Takahiro Atsumi, Hideo Enoki.

**Formal analysis:** Ayataka Fujimoto, Tohru Okanishi.

**Methodology:** Keisuke Hatano.

**Supervision:** Hideo Enoki, Tohru Okanishi.

**Writing – original draft:** Yotaro Asano, Ayataka Fujimoto.

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
