## [Decision Letter · Decision Letter 0]

23 Jan 2023

PONE-D-22-34521Non-1st seizure was less severe than 1st seizure with non-urgent level among suspected seizures transferred by ambulancePLOS ONE

Dear Dr. Fujimoto,

Thank you for submitting your manuscript to PLOS ONE. After careful consideration, we feel that it has merit but does not fully meet PLOS ONE’s publication criteria as it currently stands. Therefore, we invite you to submit a revised version of the manuscript that addresses the points raised during the review process.

We look forward to receiving your revised manuscript.

Kind regards,

Tommaso Martino, M.D.

Academic Editor

PLOS ONE

Journal Requirements:

2. In the ethics statement in the manuscript and in the online submission form, please provide additional information about the patient records/samples used in your retrospective study. Specifically, please ensure that you have discussed whether all data/samples were fully anonymized before you accessed them and/or whether the IRB or ethics committee waived the requirement for informed consent. If patients provided informed written consent to have data/samples from their medical records used in research, please include this information.

3. Thank you for stating the following financial disclosure: "No"

4. Thank you for stating the following in your Competing Interests section: "No"

6. Please amend your authorship list in your manuscript file to include author "Hideo Enoki".

Reviewers' comments:

Reviewer's Responses to Questions

**Comments to the Author**

1. Is the manuscript technically sound, and do the data support the conclusions?

Reviewer #1: Partly

Reviewer #2: Partly

Reviewer #3: Yes

Reviewer #4: No

Reviewer #5: Yes

2. Has the statistical analysis been performed appropriately and rigorously? 

Reviewer #1: No

Reviewer #2: No

Reviewer #3: Yes

Reviewer #4: No

Reviewer #5: Yes

3. Have the authors made all data underlying the findings in their manuscript fully available?

Reviewer #1: No

Reviewer #2: No

Reviewer #3: No

Reviewer #4: Yes

Reviewer #5: Yes

4. Is the manuscript presented in an intelligible fashion and written in standard English?

Reviewer #1: No

Reviewer #2: Yes

Reviewer #3: Yes

Reviewer #4: No

Reviewer #5: Yes

5. Review Comments to the Author

Reviewer #1: Authors compare patients with suspected seizures on their first transfer by ambulance and those who had been transferred ≥ 2 times. During a 5 y retrospective study In this study 636 patients were evaluated and authors conclude that seizure severity was lower in the Non-1 st Group than in the 1 st seizure Group .The subject is of interest, I have following comments

1. Abstract: Should not have introduction, instead a line or two may be given as background and state the aim of the study .

2. The conclusion in abstract is too long , should state in a line or tow

Introduction: “In Status epilepticus neuronal injury Is the main concern” in my opinion physical injury is also very important probably more important than neuronal injury

Result: The result is repetitive and repeats the findings in Table 1, 2 and 3 .I suggest the main findings should be briefly stated and referred to the respective table, .Similarly the statistical tests used in the analysis should not be re stated as they have been mentioned in statistical method section or the tests may be mentioned in respective Table as footnote

The discussion is not focussed, I would suggest that briefly summarise the major findings in a line or tow and discus the results in the light of published information /literature

It is a good study, well conducted but I would like it to be shorter to make it more readable and less repetitive.

Reviewer #2: Asano et al. present a manuscript aiming to compare the severity of seizures between patients who are transferred to the hospital by ambulance for the first seizure and patients who are transferred for a second or later seizure. The reason for asking this question was to inform guidelines about the requirement for an ambulance call after seizures depending on if it was a first or a later seizure. The authors report that first-time seizures were more severe and argue that this supports current guidelines which do not necessarily require calling an ambulance for the second or later seizures.

The data is principally interesting but seizure severity may not be the main factor that should drive the decision if an ambulance need to be called. Even if a first seizure was not severe, the fact that there may be an acute cause for the first event requiring urgent treatment suggests that first seizures should be worked up urgently. On the other hand, if the diagnosis of epilepsy is known and the seizures are typical, then there is very unlikely an acute symptomatic cause, i.e. the decision to transport to the hospital depends on event severity. Therefore, assessing severity of the second or later seizure as done in this study is reasonable but I do not see why the comparison to the severity of the first event is helpful as these need to be seen urgently anyway even if some patients do not need to be admitted to the hospital once an acute cause has been excluded. Of note, 5% of the patients presenting after the first seizure had status epilepticus or required admission to the ICU which was similar to the patients after the second or later seizure (4.8%). There was a lower percentage which required admission to the hospital overall after the second seizure, probably as the general epilepsy workup had already been completed in the past. This may not reflect "severity" as interpreted in the study.

There are also methodological issues that need to be addressed.

-The inclusion criteria are not clear to me. The authors list 4 different inclusion criteria that are combined with "or". Criterion 1 is "patients transferred due to possible CNS disorders". This would mean everybody with a neurological cause would be included which does not make sense. They later list "keywords" (page 15/16) which are not keywords but rather search criteria. But again, these do not help much to understand the selection process. Did they include patients transported to the ED AND seen by epilepsy specialists? Then they would have missed the ones that were not seen by epilepsy specialists. The authors need to very clearly state how they selected the included patients and which search criteria they used.

- The authors use an ordinal scale to describe "severity". They also comment that they used the U test as the data did not follow normal distribution. However, they calculate mean and standard deviation for this ordinal scale which does not make sense.

- It is confusing that a numbered scale is also used for the transport reasons. This scale is clearly not ordinal so it may be better to remove the numbers. Also, it appears to me that this "reason for transport" was not the reason used at the time the decision for transport was made but rather later identified. For example, PNES was defined by ED physicians (page 15) so could not have been a reason for transport. I wonder if the authors rather want to describe the cause for the transport?

Other issues:

- The authors should provide information how many of their patients overlap between the group with first seizures and the group with 2nd and later seizure.

- The severity score is used in the abstract but the reader does not know the definition.

Reviewer #3: This paper was aimed to compare severity of suspected seizures between two groups of patients, i.e those with suspected seizures transferred by ambulance for either the first time (1st Group) or at least the second time (Non-1st Group). The grade of suspected seizure severity was lower in the Non-1st Group than in the 1st Group. These findings confirm previous literature reports that ambulance transport is often unnecessary, and suggest that other strategies might be useful for patients with

suspected seizures.

Epidemiological data are described in detail, and well addressed in the Discussion. Statistical analysis is adequate.

Reviewer #4: The current data and results do not fully support the conclusion.

1) Table 4 shows significant difference in severity grade 3 and grade 4 in 1st and Non-1st Groups. How about the severity grade for the Non-1St Group at the first ambulance transport only? Is there any difference compared to those of 1st group and those of Not-1st groups at the following transports.

2) In Conclusion, authors note that "About 95% of Non-1st Group suspected seizures were more than grade 4". Most patients who are in Non-1st group are on Anti-seizure medication (ASM). Patients may be instructed to follow up with their neurologists once their conditions are stable at ER. Will this affect severity assessment in this study?

3) It is grateful if authors can present more details on how to reduce unnecessary use of ambulances on page 23.

Reviewer #5: The study is interesting and well-written. However, in my opinion there are major shortcomings to be addressed in order to be considered for publication:

INTRODUCTION

• “The reason for timing seizures is that seizures lasting >5 min can be considered status epilepticus (SE)” Please specify whether you are speaking of convulsive seizures. Otherwise, change your timeline (see “A definition and classification of status epilepticus – Report of the ILAE Task Force on Classification of Status Epilepticus” doi: 10.1111/epi.13121).

• However, the problem of inappropriate or unnecessary ambulance calls is a problem in the real world. Please double check repetitions (e.g. problem in this sentence)

METHODS

• I do not understand whether you explored only tonic-clonic seizures and mimics or included all seizure types. Should you focus on convulsive semiology, you should opt between these two options (1) convulsive seizures and all mimics (e.g. syncope, not only PNES (2) convulsive seizures only – it is a retrospective study, you can choose.

• “in need of attention from an epilepsy specialist”. How could you evaluate this criterium? As a referral after general neurological consultancy at the ED? If so, please specify how referrals are managed at your site and whether you can actually distinguish between referrals to neurologists (generally speaking) and epileptologists. Please clarify how the neurologcy emergency service is organized at your Hospital.

• “patients with psychiatric symptoms including psychogenic non-epileptic seizure (PNES)”. The diagnosis of PNES should be referred to published guidelines (see “Classification and comparative analysis of psychogenic nonepileptic seizures (PNES) semiology based on video-electroencephalography (VEEG)” doi: 10.1016/j.yebeh.2020.107697)? In my opinion it is redundant herein, this point can be included within number 1 inclusion criterium

• Exclusion criteria: “3) patients who had apparent nonepileptic, CNS disorders such as subarachnoid or intracranial hemorrhage, chronic subdural hemorrhage, CNS infection, etc.;” This collides with the first inclusion criterium. For instance, acute symptomatic seizures can result from encephalitis and non-convulsive seizures may be hardly distinguishable from other acute CNS disorders

• “We conducted comparisons using the most severe grade for the Non-1st Group, rather than mean or median in uni- and multivariate logistic regression analysis”. Please explicate the meaning of “Most severe grade”. If not better justified, mean values should be included in the analysis to draw firm conclusions.

RESULTS

• Please clarify you did not count multiple times patients in the non 1st seizure group

• Is there any difference for paediatric patients?

DISCUSSION

• “In this study, the fact that most trips by the Non-1st Group were for grades 4 and 5, and only 0.2% of trips required resuscitation and 3% required admission to the intensive care unit is also worth knowing. Since we did not study whether specific trips were unnecessary in this study, we cannot conclude that most trips for the Non-1st Group were unnecessary. However, the collected data revealed that most trips for the Non-1st Group did not require admission to hospital”. Where do these data come from? There is no mention in the results

• A discussion on possible seizure precipitants in patients with established epilepsy should be considered, as possible trigger of seizures should be taken into account when assessing the non 1st seizure group (see “ Dealing with the storm: An overview of seizure precipitants and spontaneous seizure worsening in drug-resistant epilepsy” doi: 10.1016/j.yebeh.2019.05.036)

TABLES AND FIGURES

Please add figures that may help the readership in interpreting the results (especially for the logit analysis).

6. PLOS authors have the option to publish the peer review history of their article (what does this mean?). If published, this will include your full peer review and any attached files.

Reviewer #1: No

Reviewer #2: No

Reviewer #3: No

Reviewer #4: **Yes: **Xiaoming Zhang

Reviewer #5: No

---

## [Author Response · Author response to Decision Letter 0]

15 May 2023

Response to the comments from the reviewers

We greatly appreciate the thorough review and kind advice, which have helped us to improve the quality of our manuscript. We are sorry for the delay in providing our responses to the questions raised, as we needed time to revise the text. The manuscript has been amended to clarify the issues raised by the reviewer. Our changes and point-by-point responses to the comments made are summarized below. We have also tracked the changes in red in the revised manuscript.

Reviewer #1:

Authors compare patients with suspected seizures on their first transfer by ambulance and those who had been transferred ≥ 2 times. During a 5 y retrospective study In this study 636 patients were evaluated and authors conclude that seizure severity was lower in the Non-1 st Group than in the 1 st seizure Group .The subject is of interest, I have following comments

1. Abstract: Should not have introduction, instead a line or two may be given as background and state the aim of the study .

Response:

We have amended the Abstract as suggested.

2. The conclusion in abstract is too long , should state in a line or tow

Response:

We have rephrased the suggested parts.

Introduction: “In Status epilepticus neuronal injury Is the main concern” in my opinion physical injury is also very important probably more important than neuronal injury

Response:

We have clarified this issue in the Introduction section. We have also added a phrase citing tree references in the Discussion section.

Result: The result is repetitive and repeats the findings in Table 1, 2 and 3 .I suggest the main findings should be briefly stated and referred to the respective table, .Similarly the statistical tests used in the analysis should not be re stated as they have been mentioned in statistical method section or the tests may be mentioned in respective Table as footnote

Response:

We are sorry for the redundant descriptions in the Results section. We have added subsection 3.1 Main findings in the Results section.

The discussion is not focussed, I would suggest that briefly summarise the major findings in a line or tow and discus the results in the light of published information /literature

It is a good study, well conducted but I would like it to be shorter to make it more readable and less repetitive.

Response:

We agree with these comments and have added key results.

Reviewer #2:

Asano et al. present a manuscript aiming to compare the severity of seizures between patients who are transferred to the hospital by ambulance for the first seizure and patients who are transferred for a second or later seizure. The reason for asking this question was to inform guidelines about the requirement for an ambulance call after seizures depending on if it was a first or a later seizure. The authors report that first-time seizures were more severe and argue that this supports current guidelines which do not necessarily require calling an ambulance for the second or later seizures.

The data is principally interesting but seizure severity may not be the main factor that should drive the decision if an ambulance need to be called. Even if a first seizure was not severe, the fact that there may be an acute cause for the first event requiring urgent treatment suggests that first seizures should be worked up urgently. On the other hand, if the diagnosis of epilepsy is known and the seizures are typical, then there is very unlikely an acute symptomatic cause, i.e. the decision to transport to the hospital depends on event severity. Therefore, assessing severity of the second or later seizure as done in this study is reasonable but I do not see why the comparison to the severity of the first event is helpful as these need to be seen urgently anyway even if some patients do not need to be admitted to the hospital once an acute cause has been excluded. Of note, 5% of the patients presenting after the first seizure had status epilepticus or required admission to the ICU which was similar to the patients after the second or later seizure (4.8%). There was a lower percentage which required admission to the hospital overall after the second seizure, probably as the general epilepsy workup had already been completed in the past. This may not reflect "severity" as interpreted in the study.

Response:

Since ED physicians do not perform the general workup for epilepsy, patients who were transferred to ED by ambulance are referred to general neurological physicians. Then, if those clinicians consider that the patient needs to see an epilepsy specialist, they are referred to an epilepsy specialist in our facility. We have added this information to the revised text.

There are also methodological issues that need to be addressed.

-The inclusion criteria are not clear to me. The authors list 4 different inclusion criteria that are combined with "or". Criterion 1 is "patients transferred due to possible CNS disorders". This would mean everybody with a neurological cause would be included which does not make sense. 

Response:

We agree that our initial descriptions were unclear and have rephrased the part in question.

They later list "keywords" (page 15/16) which are not keywords but rather search criteria. But again, these do not help much to understand the selection process. Did they include patients transported to the ED AND seen by epilepsy specialists? Then they would have missed the ones that were not seen by epilepsy specialists. The authors need to very clearly state how they selected the included patients and which search criteria they used.

Response:

We have changed the “keywords” to “search criteria”. For the explanation of “patients transported to the ED AND seen by epilepsy specialists” were included in criterion: 1) who were considered in need of attention from an epilepsy specialist among ED physicians. We have added some explanations about our system.

- The authors use an ordinal scale to describe "severity". They also comment that they used the U test as the data did not follow normal distribution. However, they calculate mean and standard deviation for this ordinal scale which does not make sense.

Response:

We have changed the data provided from mean and SD to median and interquartile range.

- It is confusing that a numbered scale is also used for the transport reasons. This scale is clearly not ordinal so it may be better to remove the numbers. Also, it appears to me that this "reason for transport" was not the reason used at the time the decision for transport was made but rather later identified. For example, PNES was defined by ED physicians (page 15) so could not have been a reason for transport. I wonder if the authors rather want to describe the cause for the transport?

Response:

We have removed the numbers in accordance with these comments.

We have also changed the phrase “reason for transport” to “etiology after transport”.

Other issues:

- The authors should provide information how many of their patients overlap between the group with first seizures and the group with 2nd and later seizure.

Response:

This represents an important point, but might be a different topic regarding the hypothesis in this study. We have been collecting data about unprovoked seizures and epilepsy, with the intention of writing another paper. We hope the reviewer will understand.

- The severity score is used in the abstract but the reader does not know the definition.

Response:

We have added the definition in the Abstract.

Reviewer #3: 

This paper was aimed to compare severity of suspected seizures between two groups of patients, i.e those with suspected seizures transferred by ambulance for either the first time (1st Group) or at least the second time (Non-1st Group). The grade of suspected seizure severity was lower in the Non-1st Group than in the 1st Group. These findings confirm previous literature reports that ambulance transport is often unnecessary, and suggest that other strategies might be useful for patients with suspected seizures.

Epidemiological data are described in detail, and well addressed in the Discussion. Statistical analysis is adequate.

Response:

Thank you very much for reviewing our manuscript. We appreciate your time and effort in providing advice on our paper.

Reviewer #4:

The current data and results do not fully support the conclusion.

1) Table 4 shows significant difference in severity grade 3 and grade 4 in 1st and Non-1st Groups. How about the severity grade for the Non-1St Group at the first ambulance transport only? Is there any difference compared to those of 1st group and those of Not-1st groups at the following transports.

Response:

This is an interesting point. Since the primary endpoint was a comparison between 1st and Non-1st Groups, comparisons of severity between the 1st Group and first transport of the Non-1st Group might be better discussed in another study. As described in Table 5, ASS was higher and PNES was less in the 1st Group than in the Non-1st Group. We hope this response is acceptable.

2) In Conclusion, authors note that "About 95% of Non-1st Group suspected seizures were more than grade 4". Most patients who are in Non-1st group are on Anti-seizure medication (ASM). Patients may be instructed to follow up with their neurologists once their conditions are stable at ER. Will this affect severity assessment in this study?

Response:

Thank you very much for raising this important suggestion. Since etiology after trip (Table 5) showed a frequency of definitive seizures of 67% in the 1st Group and 65% in Non-1st Group, we thought that taking ASM might not have affected severity.

3) It is grateful if authors can present more details on how to reduce unnecessary use of ambulances on page 23.

Response:

This is another important suggestion. We plan to provide a card as a simplified version of Table 6 that could provide a graphic summary that people can easily understand and ask patients with epilepsy to always carry this card with them. We have added this explanation to the Discussion section.

Reviewer #5:

The study is interesting and well-written. However, in my opinion there are major shortcomings to be addressed in order to be considered for publication:

INTRODUCTION

• “The reason for timing seizures is that seizures lasting >5 min can be considered status epilepticus (SE)” Please specify whether you are speaking of convulsive seizures. Otherwise, change your timeline (see “A definition and classification of status epilepticus – Report of the ILAE Task Force on Classification of Status Epilepticus” doi: 10.1111/epi.13121).

Response:

We meant convulsive seizures and have added “convulsive” before “seizures”.

• However, the problem of inappropriate or unnecessary ambulance calls is a problem in the real world. Please double check repetitions (e.g. problem in this sentence)

Response:

Thank you for pointing this out. We have modified the text to read: “inappropriate or unnecessary ambulance calls are a problem in the real world.”

METHODS

• I do not understand whether you explored only tonic-clonic seizures and mimics or included all seizure types. Should you focus on convulsive semiology, you should opt between these two options (1) convulsive seizures and all mimics (e.g. syncope, not only PNES (2) convulsive seizures only – it is a retrospective study, you can choose.

Response:

We have added the sentence, “We included patients not only with tonic-clonic seizures but also patients with other seizure types or mimics (psychogenic non-epileptic seizure [PNES] and syncope)”.

• “in need of attention from an epilepsy specialist”. How could you evaluate this criterium? As a referral after general neurological consultancy at the ED? If so, please specify how referrals are managed at your site and whether you can actually distinguish between referrals to neurologists (generally speaking) and epileptologists. Please clarify how the neurologcy emergency service is organized at your Hospital.

Response:

We have added our system for referrals.

• “patients with psychiatric symptoms including psychogenic non-epileptic seizure (PNES)”. The diagnosis of PNES should be referred to published guidelines (see “Classification and comparative analysis of psychogenic nonepileptic seizures (PNES) semiology based on video-electroencephalography (VEEG)” doi: 10.1016/j.yebeh.2020.107697)? In my opinion it is redundant herein, this point can be included within number 1 inclusion criterium

Response:

We have added the sentence, “The patients with PNES were diagnosed with long-term video-electroencephalography” along with the suggested reference. Since including the first inclusion criterion was difficult, we have separated this content on PNES from the statement about inclusion criteria. We hope the text is now easier to understand.

• Exclusion criteria: “3) patients who had apparent nonepileptic, CNS disorders such as subarachnoid or intracranial hemorrhage, chronic subdural hemorrhage, CNS infection, etc.;” This collides with the first inclusion criterium. For instance, acute symptomatic seizures can result from encephalitis and non-convulsive seizures may be hardly distinguishable from other acute CNS disordersf

Response:

Thank you for pointing out this overlap. As this sentence may have caused some confusion, we have rephrased the text and trust that the modified sentence is more understandable.

• “We conducted comparisons using the most severe grade for the Non-1st Group, rather than mean or median in uni- and multivariate logistic regression analysis”. Please explicate the meaning of “Most severe grade”. If not better justified, mean values should be included in the analysis to draw firm conclusions.

Response:

Thank you for pointing this out. In the Non-1st Group, some patients were transferred many times during the study period. For these patients, we investigated grade of severity for each ambulance visit and used the highest grade of severity for the purposes of comparison. We have modified the sentence as follows: “If Non-1st Group patients were transferred twice or more, we conducted comparisons using the most severe grade rather than mean or median in uni- and multivariate logistic regression analysis.”

RESULTS

• Please clarify you did not count multiple times patients in the non 1st seizure group

Response:

Thank you for this pertinent suggestion. We have added the explanation in 2.4.1. 

• Is there any difference for paediatric patients?

Response:

We did not investigate differences between the settings of pediatric and adult patients. We are therefore unable to answer this question.

DISCUSSION

• “In this study, the fact that most trips by the Non-1st Group were for grades 4 and 5, and only 0.2% of trips required resuscitation and 3% required admission to the intensive care unit is also worth knowing. Since we did not study whether specific trips were unnecessary in this study, we cannot conclude that most trips for the Non-1st Group were unnecessary. However, the collected data revealed that most trips for the Non-1st Group did not require admission to hospital”. Where do these data come from? There is no mention in the results

Response:

In both grades 4 and 5, patients did not need hospitalization. These data come from the fact that more than 95% of the Non-1st Group was classified as grade 4 or 5. We have modified the sentence as follows, “In this study, the fact that most trips by the Non-1st Group were for grades 4 and 5, which do not require admission to the hospital, and only 0.2% of trips required resuscitation.”

• A discussion on possible seizure precipitants in patients with established epilepsy should be considered, as possible trigger of seizures should be taken into account when assessing the non 1st seizure group (see “ Dealing with the storm: An overview of seizure precipitants and spontaneous seizure worsening in drug-resistant epilepsy” doi: 10.1016/j.yebeh.2019.05.036)

Response:

We have added these sentences to the Limitations: “This study did not investigate the effect of treatment changes or provocative factors such as stress, sleep deprivation, and alcohol on seizure occurrence, although these factors are reported to aggravate seizures. Studies that include these factors may be desired.”

TABLES AND FIGURES

Please add figures that may help the readership in interpreting the results (especially for the logit analysis).

Response:

If accepted for publication, we will consider making a graphic summary figure.

---

## [Decision Letter · Decision Letter 1]

18 Jun 2023

PONE-D-22-34521R1Non-1st seizure was less severe than 1st seizure with non-urgent level among suspected seizures transferred by ambulancePLOS ONE

Dear Dr. Fujimoto,

Thank you for submitting your manuscript to PLOS ONE. After careful consideration, we feel that it has merit but does not fully meet PLOS ONE’s publication criteria as it currently stands. Therefore, we invite you to submit a revised version of the manuscript that addresses the points raised during the review process.

We look forward to receiving your revised manuscript.

Kind regards,

Tommaso Martino, M.D.

Academic Editor

PLOS ONE

Reviewers' comments:

Reviewer's Responses to Questions

**Comments to the Author**

1. If the authors have adequately addressed your comments raised in a previous round of review and you feel that this manuscript is now acceptable for publication, you may indicate that here to bypass the “Comments to the Author” section, enter your conflict of interest statement in the “Confidential to Editor” section, and submit your "Accept" recommendation.

Reviewer #2: (No Response)

Reviewer #3: All comments have been addressed

Reviewer #4: All comments have been addressed

Reviewer #5: All comments have been addressed

2. Is the manuscript technically sound, and do the data support the conclusions?

Reviewer #2: No

Reviewer #3: Yes

Reviewer #4: Yes

Reviewer #5: Yes

3. Has the statistical analysis been performed appropriately and rigorously? 

Reviewer #2: No

Reviewer #3: Yes

Reviewer #4: Yes

Reviewer #5: Yes

4. Have the authors made all data underlying the findings in their manuscript fully available?

Reviewer #2: No

Reviewer #3: Yes

Reviewer #4: Yes

Reviewer #5: Yes

5. Is the manuscript presented in an intelligible fashion and written in standard English?

Reviewer #2: No

Reviewer #3: Yes

Reviewer #4: Yes

Reviewer #5: Yes

6. Review Comments to the Author

Reviewer #2: Many thanks for revising the manuscript. Some of the issues I raised were appropriately addressed. However, other points were not addressed or raised additional concerns.

I had two main concerns that were not addressed by the authors:

- "seizure severity may not be the main factor that should drive the decision if an ambulance need to be called. Even if a first seizure was not severe, the fact that there may be an acute cause for the first event requiring urgent treatment suggests that first seizures should be worked up urgently": This could be mentioned in the introduction and then a statement made that this does not apply anymore for the second seizure and that the authors are, therefore, looking at severity of the actual event.

"capturing the 4.8% patients even after a 2nd seizure who need ICU admission may be very important and therefore the transport justified!": This must be pointed out in the discussion. The authors should comment if this can be recognized by the ambulance i.e. if these all had severity 1-2. If yes, a recommendation could be made which severity score would trigger transport to the hospital with a 2nd seizure. In the current form, the manuscript gives no recommendation regarding this and the reader may (possibly appropriately) think that a transfer after a 2nd seizure is needed to capture the 4.8% who are severely ill.

The authors answered to the above two questions that patients are worked up in ED by general neurology and only if needed an epilepsy specialist is called. This is the response to a different concern I had but the two main concerns I mentioned above are not addressed.

Additional issues arising from the answers to my or the other reviewer's questions:

- A major issue is that they only looked at patients who were eventually referred to an epileptologist. Patients taken care of by neurology and not referred to an epileptologist were not included. So, if a patient was found in ED to have an intracranial bleeding causing the seizure, they could have been treated without an epileptologist being involved as the pathway was clear and there was no need to involve an epileptologist. Missing such a severe etiology would certainly bias the study. In addition, the ED physician or neurologist may not have felt the need to call the epileptologist in case of an established epilepsy. Furthermore, they excluded "patients who had already been diagnosed with an established epilepsy at other facilities and first transfer to our facility". The latter would certainly fit the criteria for patients with a second or later seizure. This means that the cohort in this study is biased and may not be appropriate to answer the question.

- The inclusion and exclusion criteria are very confusing and do not clearly state the cohort that was included in the study. For example. the exclusion criteria include "patients who had apparent non-epileptic CNS disorders such as subarachnoid or intracranial hemorrhage, chronic subdural hemorrhage, CNS infection". This is confusing. Do they mean patients who did not have seizures but these disorders? Of course these should not be included if they did not have seizures but this should then rather be handled by them not fulfilling the INCLUSION criteria. As this is a study looking at seizures, the inclusion criteria should state "patients with seizures who were 1) considered in need of attention from an epilepsy specialist among ED physicians..." etc. It is fine if they screened more patients but of course if they did not have seizures they were not included.

- "All participants provided written or verbal informed consent prior to inclusion in the study". If this is correct, the authors must have excluded any patient who did not recover or passed away as they would not have been able to provide consent. This would have introduced additional bias.

In conclusion, I am quite concerned that the cohort included in the study may not be representative of the cohort the authors would like to make recommendations for.

Reviewer #3: The Authors addressed almost ll the queries and comments of the reviewers, and I think that this paper is now improved.

Reviewer #4: All comments are addressed and acceptable. The current manuscript meets the requirement for publication.

Reviewer #5: Dear Authors,

you have successfully addressed the uncertainties of the first draft.

I suggest you to draw a figure summarizing the Results of the logit analysis.

7. PLOS authors have the option to publish the peer review history of their article (what does this mean?). If published, this will include your full peer review and any attached files.

Reviewer #2: No

Reviewer #3: No

Reviewer #4: No

Reviewer #5: No

---

## [Author Response · Author response to Decision Letter 1]

26 Jun 2023

Response to the comments from the reviewers

We would like to extend our sincere thanks for your efforts in the second round of reviews. Our gratitude to all the reviewers for their thorough and detailed evaluations cannot be overstated. We are pleased to note that Reviewers #3, #4, and #5 have given their approval to the current manuscript. However, we wish to inform you that we are contemplating further revisions to incorporate the constructive feedback from Reviewer #2. We are aware that these modifications will introduce changes to the current content, yet we trust that the insights provided by Reviewer #2 are crucial for our research. We humbly seek your understanding in this regard.

As per the feedback of Reviewer #2, we have adjusted our responses accordingly, which we detail below. We would also like to take this opportunity to express our deep appreciation for the invaluable time and intellectual contribution of all the reviewers. It is indeed the insightful feedback we have received that has refined and honed our paper scientifically, for which we convey our deepest gratitude.

I had two main concerns that were not addressed by the authors:

- "seizure severity may not be the main factor that should drive the decision if an ambulance need to be called. Even if a first seizure was not severe, the fact that there may be an acute cause for the first event requiring urgent treatment suggests that first seizures should be worked up urgently": This could be mentioned in the introduction and then a statement made that this does not apply anymore for the second seizure and that the authors are, therefore, looking at severity of the actual event.

Response: 

Thank you for your feedback. We apologize for not fully understanding the two points you highlighted. In accordance with your advice, We have added the aforementioned text to the Introduction.

"capturing the 4.8% patients even after a 2nd seizure who need ICU admission may be very important and therefore the transport justified!": This must be pointed out in the discussion. The authors should comment if this can be recognized by the ambulance i.e. if these all had severity 1-2. If yes, a recommendation could be made which severity score would trigger transport to the hospital with a 2nd seizure. In the current form, the manuscript gives no recommendation regarding this and the reader may (possibly appropriately) think that a transfer after a 2nd seizure is needed to capture the 4.8% who are severely ill.

Response: 

We have modified a part of the Discussion and incorporated your pointed suggestions.

The authors answered to the above two questions that patients are worked up in ED by general neurology and only if needed an epilepsy specialist is called. This is the response to a different concern I had but the two main concerns I mentioned above are not addressed.

Response: 

We deeply apologize for failing to respond to your two questions. As you pointed out in your comments, we have made corrections in agreement with your feedback

Additional issues arising from the answers to my or the other reviewer's questions:

- A major issue is that they only looked at patients who were eventually referred to an epileptologist. Patients taken care of by neurology and not referred to an epileptologist were not included. So, if a patient was found in ED to have an intracranial bleeding causing the seizure, they could have been treated without an epileptologist being involved as the pathway was clear and there was no need to involve an epileptologist. Missing such a severe etiology would certainly bias the study. In addition, the ED physician or neurologist may not have felt the need to call the epileptologist in case of an established epilepsy. Furthermore, they excluded "patients who had already been diagnosed with an established epilepsy at other facilities and first transfer to our facility". The latter would certainly fit the criteria for patients with a second or later seizure. This means that the cohort in this study is biased and may not be appropriate to answer the question.

Response: 

Thank you for your comments. In this study, we are looking at the difference in severity between 1st Group and Non-1st Group among patients who presented with suspected seizures. As you pointed out, even if there are seizures, severe conditions like stroke which are apparent in neuro-imaging usually have significantly different treatment paths and would typically be handed over from the ER to neurology or neurosurgery. Therefore, we decided to exclude them, assuming that they wouldn’t usually be treated as suspected seizures. We appreciate your understanding.

- The inclusion and exclusion criteria are very confusing and do not clearly state the cohort that was included in the study. For example. the exclusion criteria include "patients who had apparent non-epileptic CNS disorders such as subarachnoid or intracranial hemorrhage, chronic subdural hemorrhage, CNS infection". This is confusing. Do they mean patients who did not have seizures but these disorders? Of course these should not be included if they did not have seizures but this should then rather be handled by them not fulfilling the INCLUSION criteria. As this is a study looking at seizures, the inclusion criteria should state "patients with seizures who were 1) considered in need of attention from an epilepsy specialist among ED physicians..." etc. It is fine if they screened more patients but of course if they did not have seizures they were not included.

Response: 

Your point is indeed very understandable. In this study, we excluded cases of seizure-onset stroke in our criteria. However, even if the stroke onset was initiated by a seizure, if the emergency examinations in the ER indicate that it is an acute symptomatic seizure caused by a stroke or an infection, we believe it would not be considered a ‘suspected seizure’, and there would rarely be a need for consultation with an epilepsy specialist. This may overlap with the previous answer, but we judged these exclusion criteria to be reasonable for this study.

- "All participants provided written or verbal informed consent prior to inclusion in the study". If this is correct, the authors must have excluded any patient who did not recover or passed away as they would not have been able to provide consent. This would have introduced additional bias.

Response: 

This is an important observation, and without your suggestion, it would have gone unnoticed. We appreciate your input. With regards to this point, we have added an appropriate comment to the Method section.

In conclusion, I am quite concerned that the cohort included in the study may not be representative of the cohort the authors would like to make recommendations for.

Response: 

We might be repeating ourselves here, but this study is specifically focused on cases where epileptic seizures are suspected, not conditions presenting with acute symptomatic seizures caused by severe condition. I hope you understand that the scope of this research is not as broad as what Reviewer #2 might be concerned about. I believe from the title of this paper; readers will understand that the study is limited to suspected seizures.

On the whole, we deeply appreciate your valuable comments, which have significantly improved our manuscript.

Reviewer #3: The Authors addressed almost ll the queries and comments of the reviewers, and I think that this paper is now improved.

Response: 

Thank you for spending your valuable time on this. Thanks to your input, we were able to make many improvements. We deeply appreciate it.

Reviewer #4: All comments are addressed and acceptable. The current manuscript meets the requirement for publication.

Response: 

Thank you for spending your valuable time on this. Thanks to your input, we were able to make many improvements. We deeply appreciate it.

Reviewer #5: Dear Authors,

you have successfully addressed the uncertainties of the first draft.

I suggest you to draw a figure summarizing the Results of the logit analysis.

Response: 

Thank you for spending your valuable time on this. Thanks to your input, we were able to make many improvements. We deeply appreciate it. We currently have six tables in our manuscript, and as authors, we would prefer not to add more figures to avoid complexity. We hope for your understanding in this matter. However, if you believe a figure is necessary, we are more than willing to create one. In that case, we kindly ask for your advice.

---

## [Decision Letter · Decision Letter 2]

16 Aug 2023

Non-1st seizure was less severe than 1st seizure with non-urgent level among suspected seizures transferred by ambulance

PONE-D-22-34521R2

Dear Dr. Fujimoto,

We’re pleased to inform you that your manuscript has been judged scientifically suitable for publication and will be formally accepted for publication once it meets all outstanding technical requirements.

Kind regards,

Ryan G Wagner, MSc(Med), MBBCh, PhD

Academic Editor

PLOS ONE

Additional Editor Comments (optional):

Reviewers' comments:

Reviewer's Responses to Questions

**Comments to the Author**

1. If the authors have adequately addressed your comments raised in a previous round of review and you feel that this manuscript is now acceptable for publication, you may indicate that here to bypass the “Comments to the Author” section, enter your conflict of interest statement in the “Confidential to Editor” section, and submit your "Accept" recommendation.

Reviewer #2: All comments have been addressed

Reviewer #3: All comments have been addressed

Reviewer #4: All comments have been addressed

Reviewer #5: All comments have been addressed

2. Is the manuscript technically sound, and do the data support the conclusions?

Reviewer #2: Yes

Reviewer #3: Yes

Reviewer #4: Yes

Reviewer #5: Yes

3. Has the statistical analysis been performed appropriately and rigorously? 

Reviewer #2: Yes

Reviewer #3: Yes

Reviewer #4: Yes

Reviewer #5: Yes

4. Have the authors made all data underlying the findings in their manuscript fully available?

Reviewer #2: No

Reviewer #3: Yes

Reviewer #4: Yes

Reviewer #5: Yes

5. Is the manuscript presented in an intelligible fashion and written in standard English?

Reviewer #2: Yes

Reviewer #3: Yes

Reviewer #4: Yes

Reviewer #5: Yes

6. Review Comments to the Author

Reviewer #2: Asano et al. present a revised manuscript aiming to compare the severity of seizures between patients who are transferred to the hospital by ambulance for the first seizure and patients who are transferred for a second or later seizure. The authors have successfully addressed the issues raised. I have no further comments.

Reviewer #3: I think that the Authors addressed all the queries of the reviewers, and that this paper can be accepted for publication.

Reviewer #4: (No Response)

Reviewer #5: Thank you very much indeed for your efforts. Your study looks very interesting to me and provide the readership a useful reflection for clinical practice

7. PLOS authors have the option to publish the peer review history of their article (what does this mean?). If published, this will include your full peer review and any attached files.

Reviewer #2: No

Reviewer #3: No

Reviewer #4: No

Reviewer #5: No

---

## [Editor Report · Acceptance letter]

21 Aug 2023

PONE-D-22-34521R2 

Non-1^st^ seizure was less severe than 1^st^ seizure with non-urgent level among suspected seizures transferred by ambulance 

Dear Dr. Fujimoto:

I'm pleased to inform you that your manuscript has been deemed suitable for publication in PLOS ONE. Congratulations! Your manuscript is now with our production department. 

Kind regards, 

on behalf of

Dr. Ryan G Wagner 

Academic Editor

PLOS ONE